# Coarse-grained molecular dynamics-guided immunoinformatics to explain the binder and non-binder classification of Cytotoxic T-cell epitope for SARS-CoV-2 peptide-based vaccine discovery

Muhammad Yusuf[1,2]*, Wanda Destiarani[2], Wahyu Widayat[3], Yosua Yosua[2], Gilang Gumilar[4], Angelica Shalfani Tanudireja[1], Fauzian Giansyah Rohmatulloh[2], Farhan Azhwin Maulana[2], Umi Baroroh[2,5], Ari Hardianto[1], Rani Maharani[1], Neni Nurainy[6], Acep Riza Wijayadikusumah[6], Ryan B. Ristandi[7], Ines Irene Caterina Atmosukarto[8], Toto Subroto[1,2]

1 Faculty of Mathematics and Natural Sciences, Department of Chemistry, Universitas Padjadjaran, Bandung, West Java, Indonesia, 2 Research Center for Molecular Biotechnology and Bioinformatics, Universitas Padjadjaran, Bandung, West Java, Indonesia, 3 Faculty of Pharmacy, Pharmaceutical Biology Science, Universitas Mulawarman, Samarinda, East Kalimantan, Indonesia, 4 Research Center for Electronics, National Research and Innovation Agency Republic of Indonesia (BRIN), Bandung, West Java, Indonesia, 5 Department of Biotechnology, Indonesian School of Pharmacy, Bandung, West Java, Indonesia, 6 Department of Research and Development, PT Bio Farma, Bandung, West Java, Indonesia, 7 West Java Provincial Reference Laboratory, Bandung, West Java, Indonesia, 8 Lipotek Pty Ltd. The John Curtin School of Medical Research, The Australian National University, Canberra, Australia

* m.yusuf@unpad.ac.id

## Abstract

Epitope-based peptide vaccine can elicit T-cell immunity against SARS-CoV-2 to clear the infection. However, finding the best epitope from the whole antigen is challenging. A peptide screening using immunoinformatics usually starts from MHC-binding peptide, immunogenicity, cross-reactivity with the human proteome, to toxicity analysis. This pipeline classified the peptides into three categories, i.e., strong-, weak-, and non-binder, without incorporating the structural aspect. For this reason, the molecular detail that discriminates the binders from non-binder is interesting to be investigated. In this study, five CTL epitopes against HLA-A*02:01 were identified from the coarse-grained molecular dynamics-guided immunoinformatics screening. The strong binder showed distinctive activities from the non-binder in terms of structural and energetic properties. Furthermore, the second residue from the nonameric peptide was most important in the interaction with HLA-A*02:01. By understanding the nature of MHC-peptide interaction, we hoped to improve the chance of finding the best epitope for a peptide vaccine candidate.

Data Availability Statement: The datasets analyzed during the current study are available in

the GISAID repository with accession number EPI_ISL_529967 (https://gisaid.org/), UniProtKB repository 2021_04 release (225,619,586 sequences) (https://www.uniprot.org/release-notes/2021-11-17-release), and Protein Data Bank repository (PDB ID: 1I7U) (https://www.rcsb.org/structure/1i7u).

**Funding:** M.Y. received funding from Indonesia Endowment Fund for Education (LPDP) and Indonesian Science Fund (DIPI) through RISPRO International Collaboration [grant number: RISPRO/KI/B1/KOM/5/1986/3/2020]. APC is funded by Universitas Padjadjaran. The funders had no role in study design, data collection and analysis, decision to publish, or preparation of the manuscript.

**Competing interests:** The authors have declared that no competing interests exist.

## Introduction

The increasing variants of SARS-CoV-2 viruses raised concerns about the vaccine's effectiveness, which currently is based on the whole-inactivated virus, mRNA, adenovirus viral vector, and recombinant protein [1,2]. Furthermore, since the new variant mutations occur on the surface epitope of spike protein, they might escape from the existing spike-based vaccine-induced neutralizing antibodies [3–5]. Therefore, it is important to develop a vaccine based on the conserved epitopes, yet immunogenic, to elicit a broad immune response towards many variants of Covid-19, including Omicron and its derivatives.

A successful trial of a peptide-based Covid-19 vaccine showed broad protection against many problematic SARS-CoV-2 variants [6,7]. It is specifically designed to elicit T-cell immunity against the virus rather than the neutralizing antibody [8]. This kind of vaccine could be used as a booster to strengthen cellular immunity to clear the infection and halt the progression of the infection into a severe disease [9]. Also, it may be helpful for people who did not mount enough strong immune responses after the vaccine shot due to B-cell deficiencies [8].

Some studies suggest that people who have recovered from Covid-19 have T-cells that hunt and kill the infected cells, even in some who did not produce antibodies to the virus. Interestingly, Covid-19 specific T-cell immunity was also found in some people been not exposed to SARS-CoV-2 [10–12]. It is suggested that they had a history of infection by genetically similar common human coronaviruses [13,14].

This study is not meant to disregard the significance of B-cell epitopes in vaccine development. Instead, we choose the T-cell epitopes discovery as a case study to incorporate the molecular detail of peptide-MHC binding. Ideally, vaccines should elicit B-cell (neutralizing antibody) and T-cell responses, including CD8+ and CD4+ T-cell responses. However, in Ad26.COV2.S and BNT162b2 case studies, T-cell responses have been found to have cross-reactivity against different variants of SARS-CoV-2, including the Omicron variant, but not humoral response [15]. CD8+ T cells have been shown to contribute to protection against SARS-CoV-2, especially when antibody responses are suboptimal. Therefore, the inclusion of T cell responses in addition to antibody responses in vaccines is essential for robust protection against severe disease caused by SARS-CoV-2.

It is implied that some identical epitopes between human coronavirus and SARS-CoV-2 resulted in protective antibodies to Covid-19. It is worth noting that T-cell immunity also showed long-term protection [16,17]. T-cell vaccines might offer fast-response immune protection, clearing the virus rapidly, even before the infected person becomes symptomatic [9]. However, despite all the advantages of peptide-based vaccines, some concerns should be carefully considered when designing the peptide. Peptide epitopes alone have low immunogenicity, stability, and antigen uptake [18]. Therefore, liposome which contains danger signals such as Pam2Cys may overcome such problems [19].

On the bright side, peptide vaccines can be made by chemical synthesis, making them suitable for cost-effective production [18]. However, finding the best epitope peptide candidate from the whole antigen is challenging. A pipeline of peptide screening using immunoinformatics usually starts from MHC-binding peptide, immunogenicity, cross-reactivity with human proteome, conservancy, allergenicity, to toxicity predictions. Moreover, an immunoinformatics prediction classified the peptides into three categories: strong-binder, weak-binder, and non-binder, without incorporating the molecular binding mechanism. For this reason, the detail behind the molecular aspect that discriminates strong/weak-binder from non-binder is interesting to be studied. Major Histocompatibility Complex (MHC) is a critical protein that binds the antigenic peptide and presents it on the cell surface to be recognized by the T-cell receptor, followed by the activation and maturation of the T-cell to fight the infected cell in the

future [20]. Therefore, the mechanism of peptide-MHC binding is essential to be studied in the exploratory process of peptide vaccine design.

Molecular dynamics simulation is a powerful method to study the molecular interaction and energies between MHC and peptides [21–25]. Furthermore, coarse-grained MD (CGMD) simulations have been used in many studies to accelerate the timescale of observation at the atomic level, thus enabling us to analyze the interaction more thoroughly than the classical MD simulation. In addition, CGMD can perform several hundred times faster than the all-atom MD simulation [26]. Therefore, this study aimed to screen the MHC-binding peptide, specifically the CTL epitope, of SARS-CoV-2 using immunoinformatics and explain the molecular mechanism of the difference between strong-, weak-, and non-binder, classified by NetMHCPan, using CGMD simulation. Immunoinformatics screening using NetMHCPan and subsequent protocols were used to select peptides that bind MHC-I, are immunogenic, and are non-toxic to humans. Furthermore, CGMD simulations were expected to identify the key-specific interactions of the strong-binder with MHC. By understanding the molecular nature of MHC-peptide behavior explored by CGMD, we hoped to improve the chance of finding the best peptide for a vaccine candidate. This paper aims to enhance epitope-based vaccine development by employing molecular dynamics simulations to distinguish between true positive and false positive binder peptides. It is for the first time that immunoinformatics screening incorporates the molecular binding mechanisms of peptide-MHC interactions to validate binding affinities predicted by sequence-based predictor programs.

## Methods

### Sequences retrieval of complete genes encoding SARS-CoV-2 spike protein

The genetic sequences encoded the Spike (S) viral proteins from Wuhan clinical data corresponding to the accession number EPI_ISL_529967 deposited in the GISAID repository (https://www.gisaid.org/). Subsequently, the FASTA formatted sequences of this protein were defined as a query sequence in T-cell epitope predictions. As for the other variant of SARS-CoV-2, such as Delta and Omicron, we use the CoVariants web server (https://www.covariant.org/) that provides an overview of SARS-CoV-2 variants and mutations that are of interest. We have placed the workflow diagram of this study in Fig 1.

### Immunoinformatics-based screening of SARS-CoV-2 spike T-cells epitope peptide candidates

**Prediction of Cytotoxic T Lymphocytes cell epitope peptides.**  Identification and selection of spike epitope peptides against Cytotoxic T Lymphocyte cells (CTL) across distinctive HLA alleles of the Wuhan SARS-CoV-2 sequence were carried out using the NetMHCpan-4.1 server (http://www.cbs.dtu.dk/services/NetMHCpan-4.1/) [27,28]. All of the parameters used were default parameters. Nonameric peptide epitopes were selected. The HLA-A*02:01 supertype was used in the search algorithms. The epitope peptide candidates from NetMHCpan-4.1 were ranked according to the Eluted Ligand (EL) score, the likelihood of a peptide being an MHC ligand, and the binding level as a category of their binding strength to MHC.

**Prediction of immunogenicity.**  Immunogenicity is a characteristic property of peptide epitopes that can elicit an immune response since it is not sufficiently represented by its high binding affinity to HLA alleles. The IEDB immunogenicity tool (http://tools.iedb.org/immunogenicity/) was used to generate a list of immunogenic CTL epitopes, which is predicted based on the physicochemical properties of amino acids and their specific positions. Specifically, amino acids with large and aromatic side chains and positions 4–6 are more

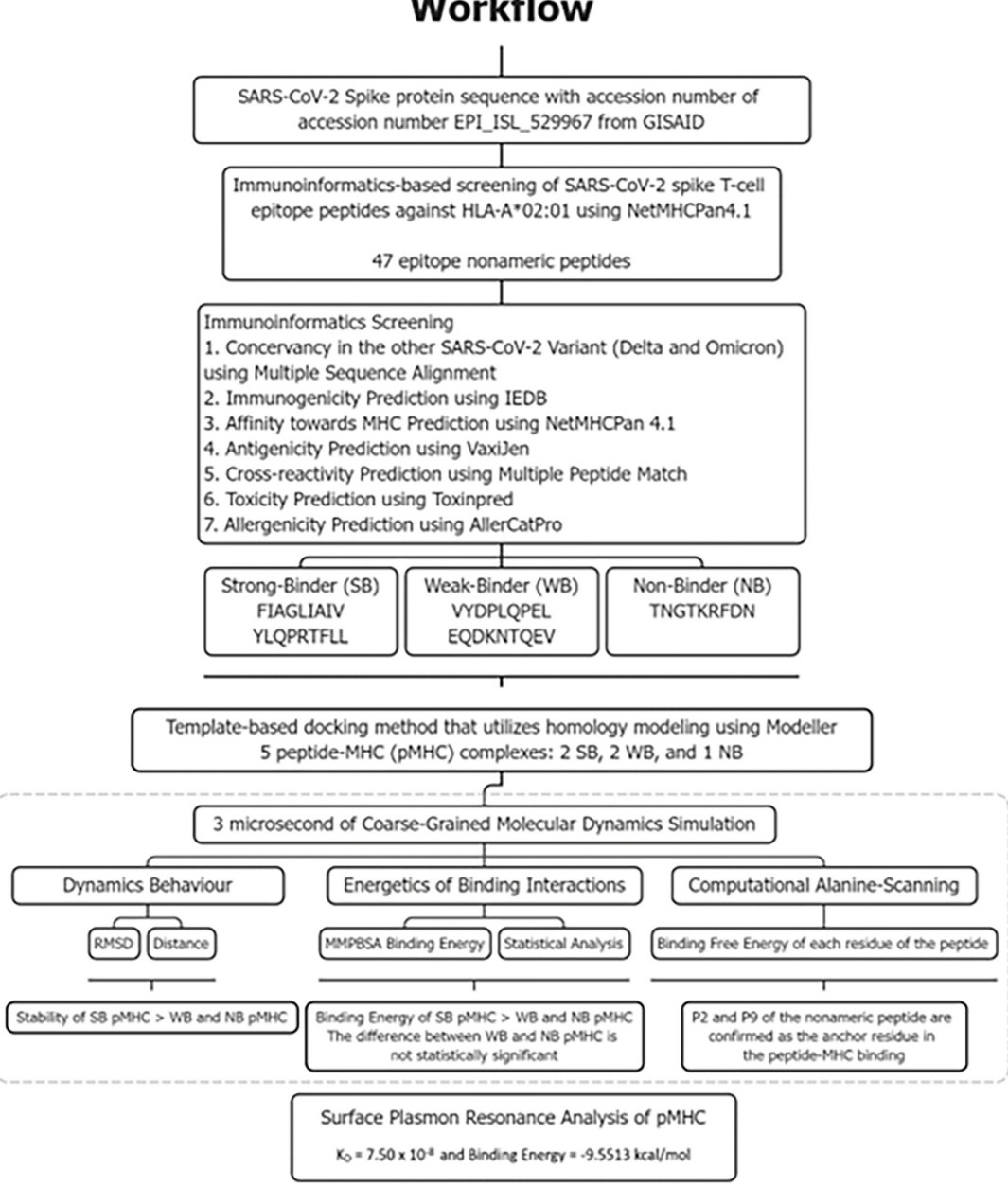

**Fig 1. The workflow.** The flow of the work done in this study, from the immunoinformatics screening to the *in silico* and *in vitro analysis* of peptide–MHC interaction, is shown in the figure.

important to the peptide's immunogenicity.[29] The ranking was conducted by sorting from lower to higher immunogenicity scores.

**Cross-reactivity and conservancy analysis.** All the immunogenic CTL epitopes obtained were used to search against human (Homo sapiens) proteome data from the UniProtKB database (2021_04 release 225,619,586 sequences as of the date 16 December 2021) for any matches to human proteome to avoid cross-reactivity. For this purpose, the Multiple Peptide Match tool (https://research.bioinformatics.udel.edu/peptidematch.jsp) of Protein Information Resource was used [30]. For the conservancy analysis, we manually aligned and matched each

of the selected epitope peptides with the complete sequences of the spike protein of SARS-CoV-2 Wuhan, Delta variant, and Omicron variant.

**Allergenicity and toxicity prediction.** AllerCatPro version 1.8 (https://allercatpro.bii.a-star.edu.sg/) was used to predict the allergenic potential of proteins based on the similarity of their 3D protein structure and their amino acid sequence. It is compared with a data set of known protein allergens comprising 4,180 unique allergenic protein sequences derived from the union of the major databases Food Allergy Research and Resource Program, Comprehensive Protein Allergen Resource, WHO / International Union of Immunological Societies, UniProtKB, and Allergome [31]. Toxinpred (https://webs.iiitd.edu.in/raghava/toxinpred.php) was also used to identify highly toxic or non-toxic peptides from many peptides submitted by a user. It predicts their toxicity and all the critical physicochemical properties, e.g., hydrophobicity, charge, and isoelectric point [32].

**Template-based docking using homology modeling.** The predictive structures of post-screening epitope peptide candidates in complex with MHC class I (HLA-A*02:01) were generated by homology modeling using the MODELLER 10.2 program [33]. The structure of the peptide-major histocompatibility complex obtained from the Protein Data Bank (PDB ID: 1I7U) was used as a reference model [34]. A few residues in the peptide, called anchor residues, bind to specific pockets on the MHC class I, resulting in some specificity of interactions with MHC, which is also used as the basis for modeling and determining the coordinates of the peptide position.

**Coarse-grained molecular dynamics (CGMD) simulations.** The 3 μs-length simulations were conducted using Amber20 for five different epitope peptide—MHC complex systems, which are obtained by immunoinformatics prediction. The SIRAH 2.0 force field was used to perform a CGMD simulation of protein in an explicit solvent [35]. In this system, explicit water (WT4) and 0.15 M of sodium and chloride ions were added to octahedral box systems. The mapping to the CG model was then applied to those systems. Two minimization stages were performed to get the lowest energy. The Langevin thermostat was used for the initial heating in the NVT ensemble, followed by an anisotropic Barendsen weak-coupling barostat up to 310K to simulate a human body temperature of about 37˚C. Position-restrained equilibration simulations were run in the NPT ensemble for 5 ns in duration, followed by the final 25 ns equilibration stage with all position restraints removed. The time step of 20 fs was used. The particle mesh Ewald method (PME) handled long-range electrostatics for protein, water, and ions interactions.

## Trajectory analysis and visualization

The RMSD and distance analysis was performed using the cpptraj program to observe the conformational structure of the peptide-MHC complex. The binding energy of the peptide-MHC complex was computed using MM/PBSA methods every 300 ns during simulations, and the interaction energy between the peptide and its closest residue within 5 Å was calculated using MM/PBSA method. Visual Molecular Dynamics (VMD) was used to visualize the peptide binding to MHC class I, while data visualizations were carried out using ggplot and matplot in Jupyter-lab. We also performed an alanine-scanning analysis on each representative conformation of systems using the FoldX plugin in the YASARA program to examine anchor residues' role in each epitope peptide candidate.

## Statistical analysis of the binding category

The statistical analysis was calculated by Dixon's Test, Shapiro-Wilk normality test, Bartlett test of homogeneity of variances, Welch's ANOVA, and Games-Howell multiple comparisons

using R in Jupyterlab. Dixon's test was used to recognize the outlier or suspicious observation in the sample [36,37]. After that, the Shapiro-Wilk test was done to examine if the binding free energy variable is normally distributed in some populations or not [38,39]. It will quantify the similarity between the observed and normal distributions as a single number, then compute which percentage of the sample overlaps with it. Bartlett's test variance assumes that variances are equal across groups or samples, which is called homogeneity of variances, and this test can be used to verify that assumption [40,41]. Welch's ANOVA is used as a major alternative to the ANOVA F test under variance heterogeneity [42,43]. Finally, the Games-Howell test was conducted to distinguish the binding level categories of the peptide [44,45].

## The pMHC stability assay

The stability of SARS-CoV-2 peptide–MHC complexes, MHC I-Strep HLA-A\*0201–YLQPRTFLL (IBA Lifesciences GmbH, Germany) were measured in vitro using Surface Plasmon Resonance (SPR). The surface of the gold chip sensor was functionalized with streptavidin (Sigma-Aldrich, Singapore) prior to the assay. First, the gold chip was immersed in piranha solution (Conc. $H_2SO_4$: 30% $H_2O_2$ = 3:1) to remove contaminants on the surface. Afterward, the gold chip was soaked in 10 mM 3-MPA overnight to functionalize it with the carboxylic group on the surface. Then, the gold chip was soaked in the EDC-NHS mixture (0.4 M EDC and 0.1 M Sulfo-NHS) for 1 hour to activate the carboxylic group. Following the activation, 50 μg/mL streptavidin (in acetic buffer pH 4.5) was immobilized by dropping the solution on the gold chip sensor's surface and incubating for 30 min. The streptavidin-coated gold chip was blocked by 1% BSA in 1X PBS pH 7.4 to prevent an unspecific binding site. Finally, the gold chip was rinsed using 1X PBS pH 7.4 on each immobilization step to remove unreacted reagents. The streptavidin-coated gold chip was assembled in Nano SPR 8 instrument (Nano SPR, USA) and equilibrated with 1X PBS pH 7.4 with a 10 μL/min flow rate. Then, 50 mM pMHC complex in 1X PBS pH 7.4 was introduced to the chip with a flow rate of 10 μL/min for 20 min. To monitor the dissociation of the pMHC complex, 1X PBS pH 7.4 continuously flowed through the chip with a flow rate of 10 μL/min for 200 min. The SPR response was recorded as sensorgram (RU/min) and analyzed using the Langmuir Adsorption Isotherm Model.

## Results

### SARS-CoV-2 spike T-cells epitope peptide candidates from immunoinformatics screening

The prediction of CTL epitope peptides against SARS-CoV-2 was performed using immunoinformatics screening with several selection steps to obtain the most potential epitope peptide. This epitope prediction was conducted using NetMHCpan1, based on its affinity to HLA-A\*02:01 as the most widespread supertype [46,47]. Further, it was sorted by the immunogenicity and affinity score parameters. This process resulted in a list of 47 nonameric peptides of SARS-CoV-2 spike protein that bind to HLA-A\*02:01 and are categorized as strong-, weak-, and non-binder (Table 1). The top two peptides are FIAGLIAIV (FIA) and YLQPRTFLL (YLQ), which had the highest immunogenicity and affinity score, labeled as the strong-binder. The EQDKNTQEV (EQD) and VYDPLQPEL (VYD) with the lowest affinity score were taken as the weak-binder. Whereas the TNGTKRFDN (TNG) was chosen as the non-binder due to its zero EL score. It is worth noting that based on the multiple-sequence alignment, these peptides were conserved among the wild type of SARS-CoV-2 and other variants of concern. Thus, suggesting broad coverage protection when these peptides are

**Table 1. The SARS-CoV-2 spike T-cells epitope peptide candidates.**

| No. | MHC | Peptide | Score EL | Affinity (nM) | Bind Level | Immunogenicity Score | Autoimmune Indication | Toxicity Prediction | Allergenicity Prediction |
|---|---|---|---|---|---|---|---|---|---|
| 1 | HLA-A*02:01 | FIAGLIAIV | 0.641405 | 6.61 | SB | 0.27206 | No | Non—Toxin | Non—Allergen |
| 2 | HLA-A*02:01 | YLQPRTFLL | 0.971198 | 4.30 | SB | 0.13050 | No | Non—Toxin | Non—Allergen |
| 3 | HLA-A*02:01 | VVFLHVTYV | 0.741670 | 17.07 | SB | 0.12780 | No | Non—Toxin | Non—Allergen |
| 4 | HLA-A*02:01 | NLNESLIDL | 0.618877 | 89.53 | SB | 0.05239 | No | Non—Toxin | Non—Allergen |
| 5 | HLA-A*02:01 | VLNDILSRL | 0.938498 | 22.77 | SB | 0.03000 | No | Non—Toxin | Non—Allergen |
| 6 | HLA-A*02:01 | GLTVLPPLL | 0.622173 | 100.50 | SB | 0.01706 | No | Non—Toxin | Non—Allergen |
| 7 | HLA-A*02:01 | RLDKVEAEV | 0.825045 | 46.73 | SB | 0.01617 | No | Non—Toxin | Non—Allergen |
| 8 | HLA-A*02:01 | RLNEVAKNL | 0.652653 | 246.13 | SB | -0.01010 | No | Non—Toxin | Non—Allergen |
| 9 | HLA-A*02:01 | KIADYNYKL | 0.864611 | 23.08 | SB | -0.10379 | No | Non—Toxin | Non—Allergen |
| 10 | HLA-A*02:01 | LLFNKVTLA | 0.803506 | 12.82 | SB | -0.11337 | No | Non—Toxin | Non—Allergen |
| 11 | HLA-A*02:01 | SIIAYTMSL | 0.580032 | 20.36 | SB | -0.12935 | No | Non—Toxin | Non—Allergen |
| 12 | HLA-A*02:01 | ALNTLVKQL | 0.657403 | 563.85 | SB | -0.18466 | No | Non—Toxin | Non—Allergen |
| 13 | HLA-A*02:01 | VLYENQKLI | 0.495902 | 359.26 | SB | -0.20427 | No | Non—Toxin | Non—Allergen |
| 14 | HLA-A*02:01 | RLQSLQTYV | 0.873760 | 11.92 | SB | -0.29331 | No | Non—Toxin | Non—Allergen |
| 15 | HLA-A*02:01 | HLMSFPQSA | 0.798454 | 41.79 | SB | -0.31433 | No | Non—Toxin | Non—Allergen |
| 16 | HLA-A*02:01 | TLDSKTQSL | 0.914998 | 175.75 | SB | -0.52715 | No | Non—Toxin | Non—Allergen |
| 17 | HLA-A*02:01 | VTWFHAIHV | 0.221377 | 134.02 | WB | 0.38925 | No | Non—Toxin | Non—Allergen |
| 18 | HLA-A*02:01 | SVTTEILPV | 0.216735 | 147.18 | WB | 0.25860 | No | Non—Toxin | Non—Allergen |
| 19 | HLA-A*02:01 | ALLAGTITS | 0.112596 | 729.27 | WB | 0.24031 | No | Non—Toxin | Non—Allergen |
| 20 | HLA-A*02:01 | KLPDDFTGC | 0.334903 | 450.41 | WB | 0.20308 | No | Non—Toxin | Non—Allergen |
| 21 | HLA-A*02:01 | NTQEVFAQV | 0.227439 | 497.10 | WB | 0.17889 | No | Non—Toxin | Non—Allergen |
| 22 | HLA-A*02:01 | VLSFELLHA | 0.350734 | 118.62 | WB | 0.16070 | No | Non—Toxin | Non—Allergen |
| 23 | HLA-A*02:01 | PLVDLPIGI | 0.115844 | 1130.35 | WB | 0.14660 | No | Non—Toxin | Non—Allergen |
| 24 | HLA-A*02:01 | YQPYRVVVL | 0.189014 | 1377.75 | WB | 0.14090 | No | Non—Toxin | Non—Allergen |
| 25 | HLA-A*02:01 | QLNRALTGI | 0.093590 | 513.78 | WB | 0.13020 | No | Non—Toxin | Non—Allergen |
| 26 | HLA-A*02:01 | FCNDPFLGV | 0.213279 | 555.89 | WB | 0.13006 | No | Non—Toxin | Non—Allergen |
| 27 | HLA-A*02:01 | FLHVTYVPA | 0.098767 | 34.61 | WB | 0.11472 | No | Non—Toxin | Non—Allergen |
| 28 | HLA-A*02:01 | ELLHAPATV | 0.336419 | 386.28 | WB | 0.11231 | No | Non—Toxin | Non—Allergen |
| 29 | HLA-A*02:01 | FQFCNDPFL | 0.369084 | 10.54 | WB | 0.05737 | No | Non—Toxin | Non—Allergen |
| 30 | HLA-A*02:01 | YTNSFTRGV | 0.121645 | 328.00 | WB | 0.04545 | No | Non—Toxin | Non—Allergen |
| 31 | HLA-A*02:01 | FTISVTTEI | 0.375875 | 58.89 | WB | 0.04473 | No | Non—Toxin | Non—Allergen |
| 32 | HLA-A*02:01 | ILDITPCSF | 0.131257 | 2524.42 | WB | 0.02632 | No | Non—Toxin | Non—Allergen |
| 33 | HLA-A*02:01 | LQIPFAMQM | 0.157345 | 863.81 | WB | -0.03301 | No | Non—Toxin | Non—Allergen |
| 34 | HLA-A*02:01 | GINASVVNI | 0.137188 | 1383.16 | WB | -0.05391 | No | Non—Toxin | Non—Allergen |
| 35 | HLA-A*02:01 | LLALHRSYL | 0.095947 | 353.28 | WB | -0.06002 | No | Non—Toxin | Non—Allergen |
| 36 | HLA-A*02:01 | VYDPLQPEL | 0.196623 | 8641.52 | WB | -0.07466 | No | Non—Toxin | Non—Allergen |
| 37 | HLA-A*02:01 | VLYQGVNCT | 0.160345 | 733.09 | WB | -0.08286 | No | Non—Toxin | Non—Allergen |
| 38 | HLA-A*02:01 | YVTQQLIRA | 0.091708 | 3246.41 | WB | -0.08464 | No | Non—Toxin | Non—Allergen |
| 39 | HLA-A*02:01 | ELDSFKEEL | 0.148970 | 4841.91 | WB | -0.10527 | No | Non—Toxin | Non—Allergen |
| 40 | HLA-A*02:01 | LITGRLQSL | 0.098152 | 3501.50 | WB | -0.10776 | No | Non—Toxin | Non—Allergen |
| 41 | HLA-A*02:01 | KQIYKTPPI | 0.171701 | 118.46 | WB | -0.14982 | No | Non—Toxin | Non—Allergen |
| 42 | HLA-A*02:01 | MIAQYTSAL | 0.176384 | 105.71 | WB | -0.18768 | No | Non—Toxin | Non—Allergen |
| 43 | HLA-A*02:01 | EQDKNTQEV | 0.112759 | 8085.04 | WB | -0.21882 | No | Non—Toxin | Non—Allergen |
| 44 | HLA-A*02:01 | RVYSTGSNV | 0.164307 | 1163.21 | WB | -0.24137 | No | Non—Toxin | Non—Allergen |
| 45 | HLA-A*02:01 | SLSSTASAL | 0.243351 | 435.97 | WB | -0.26230 | No | Non—Toxin | Non—Allergen |
| 46 | HLA-A*02:01 | ALGKLQDVV | 0.173341 | 852.18 | WB | -0.28300 | No | Non—Toxin | Non—Allergen |

*(Continued)*

**Table 1.** (Continued)

| No. | MHC | Peptide | Score EL | Affinity (nM) | Bind Level | Immunogenicity Score | Autoimmune Indication | Toxicity Prediction | Allergenicity Prediction |
|---|---|---|---|---|---|---|---|---|---|
| 47 | HLA-A*02:01 | KIYSKHTPI | 0.161841 | 463.64 | WB | -0.32094 | No | Non—Toxin | Non—Allergen |

The SARS-CoV-2 spike T-cells epitope peptide candidates against HLA-A*02:01 based on the immunoinformatic screening. Peptides selected for strong-, weak-, and non-binder are highlighted in bold.

*SB: Strong Binder; WB: Weak Binder.

developed as a vaccine. Moreover, these peptides also pass the immunoinformatics screening using the predictions of cross-reactivity with the human proteome, conservancy, allergenicity, and toxicity. Nevertheless, this study is limited to including Wild-Type, Omicron, and Delta variants. The recent XBB variant has yet to be added to the conservancy analysis. Furthermore, the selected peptides were modeled into peptide-MHC complex and simulated using CGMD to investigate the molecular aspect behind the binder and non-binder classification.

### Dynamics behavior of binder and non-binder peptides with MHC

We performed root-mean-square deviation (RMSD) and distance analysis from the three microseconds of CGMD trajectories. The RMSD data was presented by density plot to show the deviation of structure from its initial state, so it can be used to compare changes or shifts in protein conformation. It is revealed that MHC and strong-binder FIA and YLQ systems have lower average RMSD values, about 7.3 and 6.8 Å, respectively. Moreover, the RMSD histogram of the MHC and strong-binder peptide synchronized in the distance less than 5 Å regions along the simulations, indicating the perseverance of the binding between the peptides and the MHC's antigen binding groove residues (Fig 2). Although the weak-binder VYD displays a similarly low average RMSD value with strong-binder (i.e., 6.4 Å), the most populous RMSD density is higher than 5 Å. Another weak-binder EQD reached 9.5 Å while the non-binder TNG was 8.8 Å. This result indicated that the MHC binding with strong-binder FIA-YLQ is more favorable than the weak-binder VYD-EQD and non-binder TNG.

Distance analysis was conducted using the cpptraj program on Ambertools21 to show the average distance between the center of mass at the MHC and the peptides (Fig 3). The distance between the two molecules during the 3 μs of simulations is proportional to the stability of the binding. It is shown that both strong-binder peptides have a stable average distance,

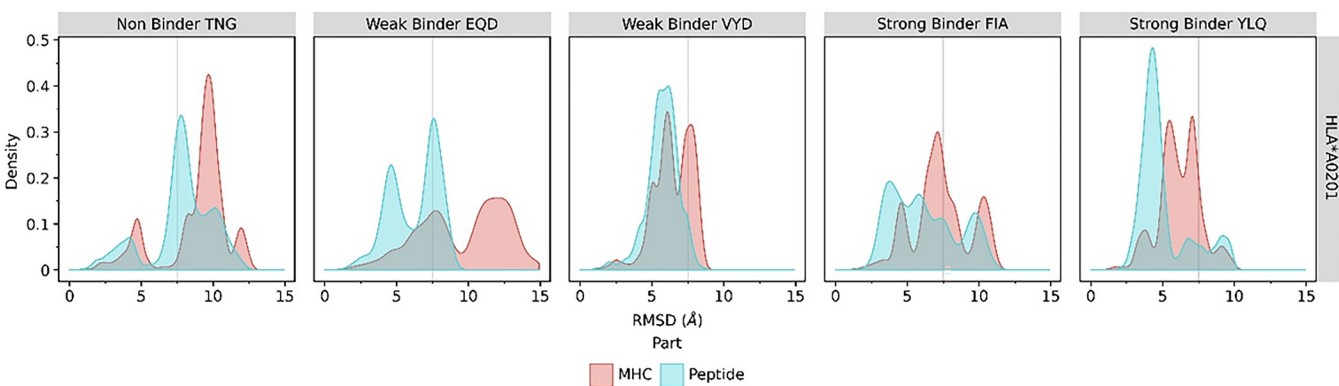

**Fig 2. The RMSD analysis.** The RMSD of MHC class I (HLA-A*02:01) and its epitope peptides from immunoinformatic screening; non-binder, weak-binder, and strong-binder as a histogram data showed the distribution of each RMSD value in all three CGMD systems during 3 μs of simulation.

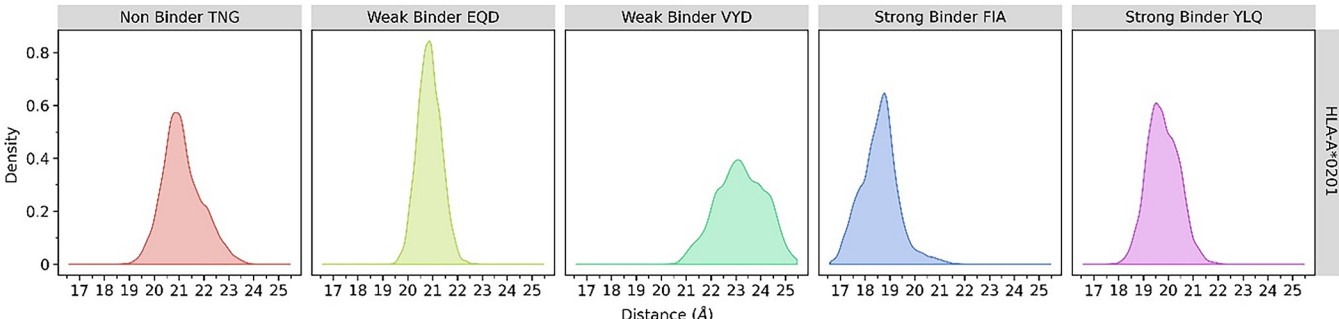

**Fig 3. The distance analysis.** The distance parameter between the center of mass of MHC class I (HLA-A*02:01) and its epitope peptides from immunoinformatic screening during three μs of coarse-grained molecular dynamics simulations.

represented by high peak densities of around 18.5 and 19.5 Å. The weak-binder EQD and non-binder TNG peptides had a farther distance, approximately 21 Å, while that of the weak-binder VYD is about 23 Å with a broader deviation range.

## Energetics of binding interactions between T-cell epitope peptides and MHC

The binding energy between the MHC and each peptide was calculated using the MMPBSA method at 300 ns intervals. The strong-binder peptides consistently showed a negative binding energy value, indicating a favorable binding. In contrast, the weak- and non-binder presented a positive value (Fig 4). The non-binder peptide has the highest binding energy that reaches +20 kcal/mol, while the strong-binder FIA has the lowest at around -16 kcal/mol. Therefore, these results are consistent with the RMSD and distance analysis (Figs 1 and 2). The strong-binder peptides have the strongest binding energy against MHC, followed by the weak-binder and non-binder, which is interestingly in line with the binding classification predicted by immunoinformatics.

Fig 4 suggested that the non-binder is qualitatively different from the weak-binder in terms of their binding energy values, while that of strong-binder was clearly different with both non- and weak-binder. Furthermore, we conducted an inferential statistics procedure to clarify our findings. The outlier in the MMPBSA values for every group was analyzed using the box-plot method. However, four potential outliers from the weak-binder were not confirmed by Dixon's test (Table 2). The normality of binding energy values for each peptide was checked using the Shapiro-Wilk test. The test showed that MMPBSA values of all peptides are normally distributed (all p-values are above 0.05). Since the Bartlett test suggested that the variance among the

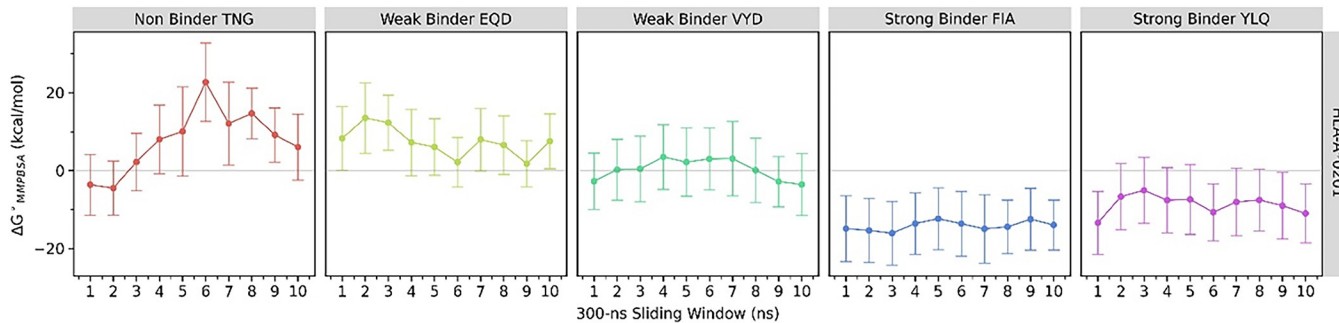

**Fig 4. The binding free energy analysis.** The binding free energy of MHC class I (HLA-A*02:01) and the epitope peptides from immunoinformatic screening were calculated by 300 ns intervals using MMPBSA methods. The grey line shows a zero value of energy.

**Table 2. The statistical analysis of binder and non-binder peptide classifications.**

| The Dixon's Test | | | |
|---|---|---|---|
| **Data** | **Q** | | **p-value** |
| WB-EQD | 0.10531 | | 0.6413 |

| The Shapiro–Wilk Test | | |
|---|---|---|
| **Systems** | **Coefficient of concordance (W)** | **p-value** |
| NB–TNG | 0.96543 | 0.8455 |
| WB–EQD | 0.92719 | 0.4208 |
| WB–VYD | 0.89537 | 0.1947 |
| SB–FIA | 0.96164 | 0.8044 |
| SB–YLQ | 0.94666 | 0.6292 |

| The Bartlett Test | | |
|---|---|---|
| **K-squared** | **Degree of freedom** | **p-value** |
| 33.091 | 4 | 0.000001144 |

| Welch's ANOVA | | | | |
|---|---|---|---|---|
| **n** | **Statistic** | **Degree of Freedom in Numerator** | **Degree of Freedom in Denominator** | **p-value** |
| 50 | 123.4 | 4 | 20.87621 | 3.27e-14 |

| The Games-Howell Test | | | | | | |
|---|---|---|---|---|---|---|
| **Group 1** | **Group 2** | **Estimate** | **Lower bound confidence interval** | **Higher bound confidence interval** | **p-adjusted** | **p-adjusted significance** |
| NB-TNG | SB-FIA | -21.86083 | -30.633795 | -13.087865 | 9.40e-05 | **** |
| NB-TNG | SB-YLQ | -16.32350 | -25.171493 | -7.475507 | 7.77e-04 | *** |
| NB-TNG | WB-EQD | -0.35021 | -9.396270 | 8.695850 | 1.00e+00 | ns |
| NB-TNG | WB-VYD | -7.34352 | -16.212933 | 1.525893 | 1.21e-01 | ns |
| SB-FIA | SB-YLQ | 5.53733 | 2.830522 | 8.244138 | 1.74e-04 | *** |
| SB-FIA | WB-EQD | 21.51062 | 17.514984 | 25.506256 | 2.39e-08 | **** |
| SB-FIA | WB-VYD | 14.51731 | 11.613838 | 17.420782 | 9.49e-09 | **** |
| SB-YLQ | WB-EQD | 15.97329 | 11.656552 | 20.290028 | 5.37e-08 | **** |
| SB-YLQ | WB-VYD | 8.97998 | 5.537714 | 12.422246 | 2.79e-06 | **** |
| WB-EQD | WB-VYD | -6.99331 | -11.396393 | -2.590227 | 1.00e-03 | *** |

The statistical analysis of the binding category in classifying T-cell peptide epitope candidates from Immunoinformatic screening.

*SB: Strong Binder; WB: Weak Binder; NB: Non Binder.

MMPBSA values is not homogenous, then the Welch analysis of variance (ANOVA) was used. It is shown that MMPBSA values among the three binding categories are statistically different (p-value = 3.27e-14). Furthermore, the post-hoc step using the Games-Howell test supports our finding that the strong-binder significantly differed with both non and weak-binders. Meanwhile, the difference in MMPBSA values between non- and weak-binders is not statistically significant (Fig 5). Despite the meaningful classification of binder and non-binder from an energetics point of view, it still lacks detail at the molecular level. Therefore, further investigation on the residual interactions between peptides and MHC is conducted to explore the key or anchor residues of the peptides that determine the strong binding.

## Computational alanine-scanning analysis of T-cell epitope peptides

The Alanine-scanning principle is to mutate every amino acid to alanine without moving the neighbors and produce a file with the resulting free Gibbs energies of alanine mutation. It is

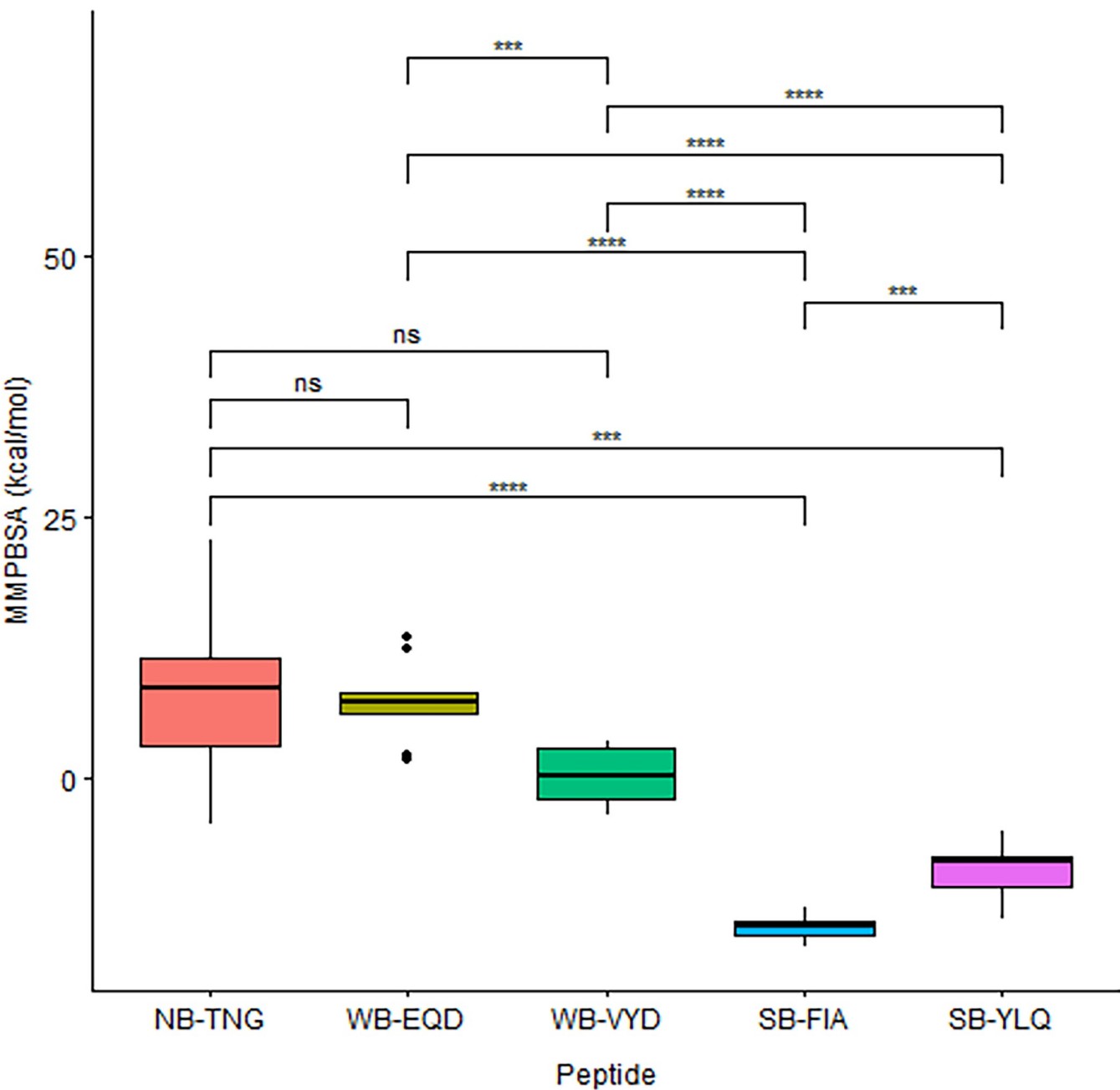

**Fig 5. The differences of statistical significance for each binding category.** The statistical graph of the binding category of peptide classification is based on the MMPBSA energy. The asterisk symbol shows the significant level of difference in each category, and the "ns" represent the non-significant statement.

helpful for a fast scan, given that the anchor residues have an essential role in the binding of the peptide to MHC class I (HLA-A*02:01). Based on the results, T-cell epitope candidates from immunoinformatics screening showed that most of the anchor residues in both strong-binder FIA and YLQ have a positive value, which means alanine mutation destabilizes the interaction between the peptide and MHC. In contrast, the alanine mutation in weak-binder EQD, VYD, and non-binder TNG resulted in negative free Gibbs energy, indicating that the original amino acid was less favorable than the alanine interaction with the MHC (Fig 6).

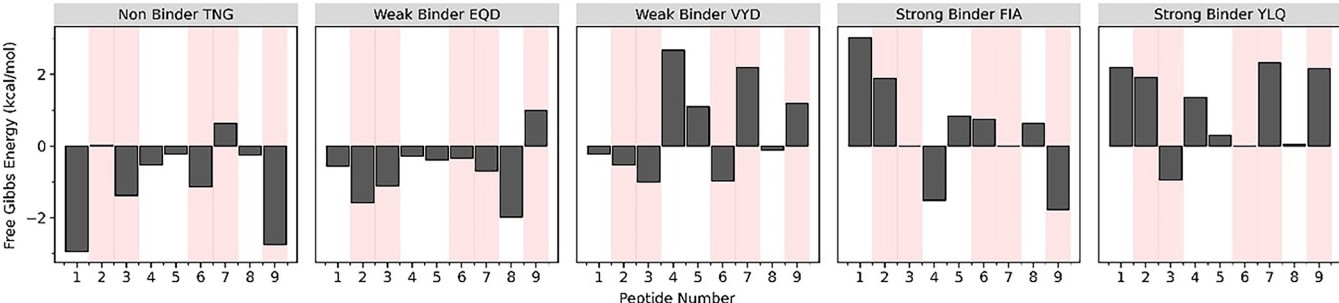

**Fig 6. The alanine-scanning analysis.** The alanine-scanning analysis of the T-cell epitope peptide's structure was extracted from the stable conformation in a frame of the 2-μs simulation. The negative value indicated the favorability of alanine mutation, while the positive is not. The red area showed the anchor residue position in each nonameric peptide.

Moreover, residue-2 was also revealed as the determinant anchor residue in categorizing the binding level of the peptides. It is shown in the alanine scanning analysis that either only residue-2 or both with residue-9 resulted in positive energy value in strong-binder peptides. Whereas in the weak-binder, only residue-9 has a positive value but not residue-2. As for the non-binder, both residues 2 and 9 have negative energy values. Therefore, residues 2 and 9 were suggested to affect the peptide binding capability towards MHC. This result is consistent with previous studies; those primary anchor residues are only presented by residues 2 and 9, while residues 3, 6, and 7 are as auxiliary or secondary [48,49]. It is noted that the other residues 1, 4, 5, and 8 were known as suboptimal residues in the peptide-MHC interaction. Instead of interacting with the MHC, the suboptimal residues directly interact with the T-cell receptor during the antigen-presenting process [48,50,51].

The residual-scale analysis on the interactions between the epitope peptides and HLA-A*02:01 suggested the contribution of each amino acid property in the peptide sequence. In general, the number of interactions between the MHC and strong-binder peptides was higher than that of weak- and non-binder peptides. Moreover, the interactions between non-binder and strong-binder to HLA-A*02:01 are different, which is presented in their binding site. The complementarity between MHC and peptide was not shown in non-binder because both residue-2 and residue-9 are the polar asparagine. In contrast, the strong-binder showed a good binding with MHC due to the presence of isoleucine at residue-2 to fit in the B-pocket and valine at residue-9 to fit in the F-pocket (Fig 7A and 7B). The distinctive MMPBSA energy of the residue-2 in strong-binder compared to weak- and non- binders is shown in Fig 7D. These results suggest that the specificity factor of primary anchor residue-2 and 9 can contribute significantly to the residual interactions with the MHC, consistent with the previous study [48].

Furthermore, the anchor residue-2 showed a better role than residue-9 in differentiating binding categories; the strong-binder FIA has the lowest interaction energy, followed by strong-binder YLQ, then weak-binder VYD and EQD. Asparagine at the second residue of non-binder TNG showed positive binding free energy values, indicating that polar moiety is not compatible with the hydrophobic B-pocket of MHC-I. On the other hand, the non-polar isoleucine and leucine from the strong-binder peptides promote the hydrophobic interaction, similar to tryptophan in weak-binder VYD. However, since tryptophan in VYD also forms electrostatic interactions with the B-pocket, it decreases the hydrophobic forces between them. This phenomenon was also observed in weak-binder EQD, where glutamine at the second residue also facilitates electrostatic interaction and hydrogen bonds. Thus, it decreased the binding affinity between EQD and the hydrophobic site of the MHC-I, close to zero (Fig 7D).

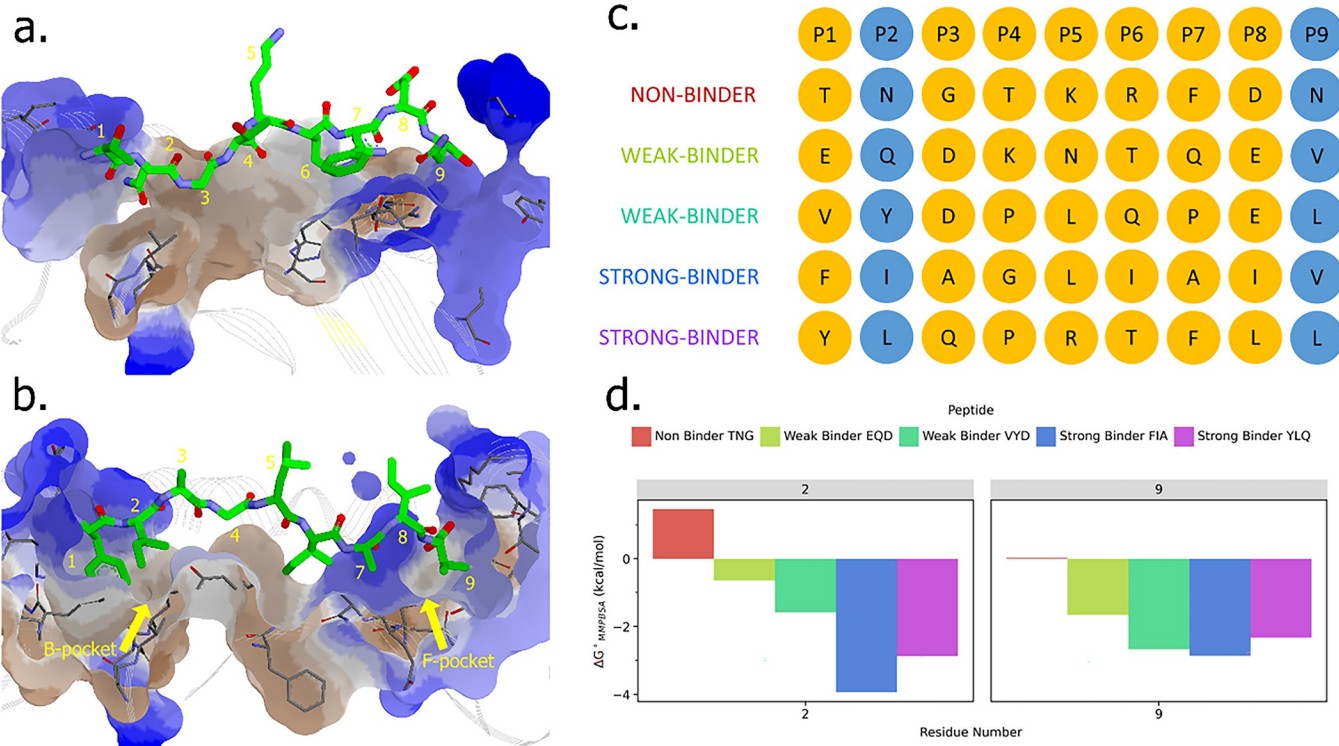

**Fig 7. The molecular detail of peptide-MHC binding mechanism.** The interactions of each T-cell epitope peptide with HLA-A*02:01; (a) non-binder and (b) strong-binder peptide. (c) The sequence of all the nonameric epitope peptides and their position. (d) The binding free energy decomposition analysis of residue-2 and 9 was calculated using the MMPBSA method.

The strong-binders and weak-binders have the same hydrophobic residue-9, i.e., leucine and valine. It facilitated the hydrophobic interaction that retains the binding of peptide and MHC. Conversely, the non-binder has a polar asparagine which is incompatible with the hydrophobic F-pocket of MHC. This analysis is consistent with the binding free energy decomposition calculations in Fig 7D.

## SPR analysis confirmation of strong-binder YLQ peptide

SPR measurements are used to determine the stability or ability of peptides to bind by pMHC. YLQ peptide bound by pMHC flowed into the binding chamber above the SPR gold plate, which had previously been immobilized with streptavidin. The binding of pMHC-peptide by streptavidin showed a response, as shown in Fig 8A. In determining its performance, an adsorption kinetics analysis was used using Anabel 2.3 software with the regression results shown in Fig 8B and 8C and the results are shown in Table 3.

From the adsorption kinetics analysis results, the pMHC-YLQ association rate ($k_{ass}$) obtained was 22247.7 $M^{-1}$. Then, to investigate the strength of the peptide bond with pMHC in terms of its dissociation response. The pMHC-YLQ dissociation rate ($k_{diss}$) obtained was YLQ = 0.001662 $s^{-1}$. The value of the affinity constant ($K_D$) also corresponds to the results above, i.e., 7.50 x $10^{-8}$ M. The smaller the $K_D$ value, the stronger the bond will be. The binding energy can be calculated using the equation $\Delta G = RT \, Ln \, K_D$ where R is the universal gas constant, and T is the room temperature in Kelvin. The calculations show that the pMHC-YLQ binding energy is -9.5513 kcal/mol. The stability was then observed through residence time ($t_{1/2}$), and the value obtained was 416.9675 s.

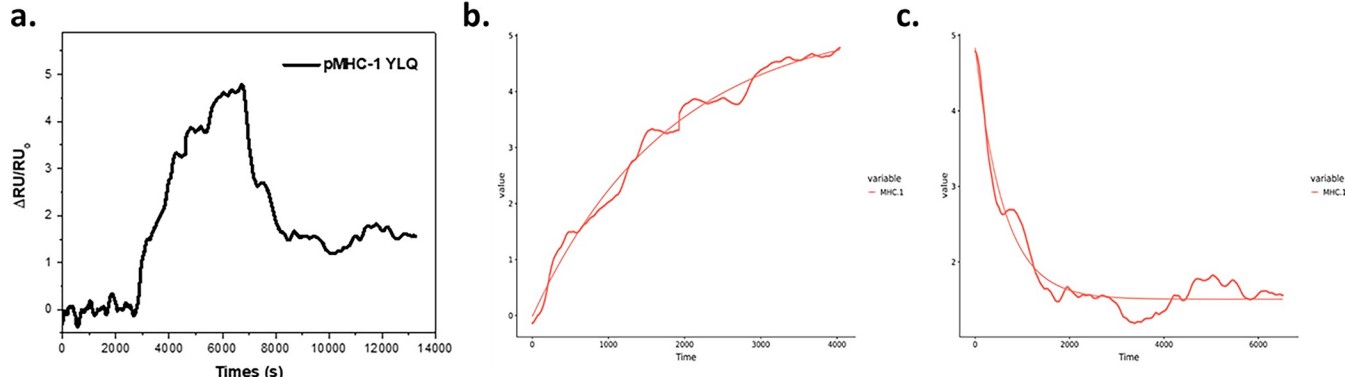

**Fig 8. The kinetics analysis of peptide-MHC binding.** The SPR sensorgram of pMHC-peptide binding by streptavidin and peptide release from pMHC(a). Kinetic analysis of pMHC-YLQ SPR response dynamic using Anabel 2.3 software; association (b) and dissociation (c).

This finding is consistent with the binding energy calculation on their residual interactions. Experimental analysis on the strong-binder YLQPRTFLL using nanoSPR displayed a high affinity towards HLA-A*02:01. For the first time, YLQPRTFLL was systematically studied from immunoinformatics screening, molecular dynamics simulation, computational binding energy, to experimental binding energy with MHC using SPR. Altogether, this pipeline proposes YLQPRTFLL as a good candidate for CTL epitope for peptide-based vaccine of SARS-CoV-2. A similar approach can be utilized to improve the chance of discovering the best epitope as a vaccine candidate. Our pipeline from immunoinformatics to CGMD and SPR analysis presented a systematic method for improving the possibility of finding the best CTL epitope candidates in vaccine development by incorporating the structural aspect of the prediction.

## Discussion

One of the vaccine types that is interesting to study is peptide-based vaccines because of their broad spectrum against multiple variants and their capability to induce cellular and humoral immunity with the least risk of allergy and autoimmune to develop [18,52]. It also presented unique properties regarding selectivity and specificity toward specific targets, making it safe and stable [6,53]. This recent year, some researchers have been focusing on this platform, as

**Table 3. The binding kinetic analysis of peptide–MHC interactions using NanoSPR.**

| Sample | YLQ-MHC I complex |
| --- | --- |
| c(Reagent) [M] | $5.0 \times 10^{-8}$ |
| $k_{obs}$ | 0.00055 |
| StErr [$k_{obs}$] | $2.27 \times 10^{-6}$ |
| $k_{diss}$ | 0.001662 |
| StErr [$k_{diss}$] | $7.72 \times 10^{-6}$ |
| $k_{ass}$ [1/M] | 22247.7 |
| StErr ($k_{ass}$) [1/M] | 160.9894 |
| $K_D$ [M] | $7.50 \times 10^{-8}$ |
| StErr ($K_D$) [M] | $6.40 \times 10^{-10}$ |
| Binding Energy (kcal/mol) | -9.5513 |
| t1/2 (Retention Time) (s) | 416.9675 |

shown by the number of peptide-based vaccines entering clinical trial phases. There are 178 vaccines in trial phase I and 115 in phase II until 2020, covering various diseases [6]. Referring to World Health Organization (WHO) reports on the Covid-19 vaccine tracker and landscape, 13 peptide-based vaccines proceed to the preclinical phase. Also, five other candidates (CoVepiT, EpiVacCorona, IMP CoVac-1, PepGNP-SARSCoV2, and UB-612) continued in the clinical phase until April 2022 [7]. For instance; CoVac-1 is a multi-peptide-based vaccine candidate designed to produce widespread and long-term SARS-CoV-2 T-cell immunity similar to that acquired by natural infection, unaffected by the variants of concern (VOCs). Furthermore, it has a decent safety profile and generates robust T-cell responses following a single immunization, according to trial phase I [8]. Hence, in this study, we project the peptide-based vaccine as our platform for the postliminary work.

Our work is restricted to the T-cell epitope candidates since the T-cell vaccines are a preferable source of top-up immunity, notably given that T-cell immunity against SARS-CoV-2 appears to last longer than antibody-mediated responses [9,54]. Interestingly, in the case of SARS-CoV and MERS-CoV, the T-cell responses against HCoVs are produced despite the moderate intensity and low frequency in the elderly [54–56]. Moreover, B-cell responses after SARS-CoV infection are typically short-lived and frequently untraceable within four years. In contrast, even after 17 years, T-cell responses can still be evoked [16,17]. As for MERS, it appears to be more powerful and persistent than humoral immunity [57]. To summarize, when B-cell responses are weak, T-cells appear capable of resolving the infection. T-cell responses specific to SARS-CoV-2 are required for viral clearance and might prevent further infection without seroconversion, offer long-term memory, and facilitate viral variant detection [17,58,59].

Discovering SARS-CoV-2 peptides that elicit T-cell responses experimentally is challenging due to the vast number of available options to test and the significant genetic diversity of MHC genes encoded HLA molecules [58]. Thus, prominent tools such as immunoinformatics were fully explored to accelerate the process and afford promising results. Equally important, this method will utterly reduce the cost as well as the required time of the research and development phase [60,61]. The definite protocols for immunoinformatics are not strictly determined; they will be different from one another and constantly improve to extend and ensure the validity of the outputs [62–65]. The immunoinformatics method involves several screening steps to obtain the most potential T-cell epitope candidates.

In this paper, about 2034 T-cell epitopes, including 416 unique sequences from the S protein of Wuhan SARS-CoV-2, were generated using The NetMHCPan4.1. Many experiments are being conducted to design an effective vaccine against SARS-CoV-2. In several of those, the S protein is considered a reasonable target for the SARS-CoV-2 vaccine since it involves viral binding, fusion, and entrance into the host cell [66,67]. Moreover, its antigenicity is well-proven by the commercial vaccine [68,69]. Further, screening parameters such as the Eluted Ligand (EL) score, affinity, immunogenicity, bind level, and conservancy parameters were used to estimate the epitope's effectiveness. The term immunogenicity refers to the ability of a substance to induce a cellular and humoral immune response, while antigenicity is the ability to be specifically recognized by the antibodies generated as a result of the immune response to the given substance [70]. At the same time, the cross-reactivity (autoimmune indication), allergenicity, and toxicity were evaluated to assess the safety standard. This process was done based on the 33 supertypes of the HLA population worldwide. The promiscuous epitope candidates were selected based on their binding with HLA-A*02:01, with the highest prevalence among other supertypes [71,72].

Around 47 T-cell epitope candidates (Table 1) were classified as strong-binder and weak-binder peptides towards HLA-A*02:01. The NetMHCPan4.1 program predicts and

discriminates T-cell epitope candidates into three bind levels or binding categories; non-binder, weak-binder, and strong-binder. The top two on the list are strong-binder FIA and YLQ. Interestingly, both show the differences between each binding category in the almost all assessment parameters from the trajectory of CGMD, such as RMSD, distance, and MMPBSA energy (Figs 2–4). Although some of these epitopes have been discovered in previous studies [73,74], this paper, for the first time, explains the molecular detail of MHC-peptides interactions. We also confirm that based on the statistical analysis of its energy, the non- and weak-binder categories are quite similar, while the strong-binder is significantly different from the other categories (Fig 5). In addition, the alanine-scanning analysis also presented that the anchor residues in strong-binder peptides are essential for the binding to HLA-A*02:01. On the contrary, the alanine mutation in the several anchor residues of non- and weak-binder is not very influential (Fig 6). The significance of the anchor residue of the peptides in its binding with MHC is widely known and studied [51,75,76]. The pockets B and F in MHC-I are critical for peptide recognition and correlated to the binding region of the N- and C-terminus of the peptide, respectively. These two pockets are responsible for accommodating specific anchor residues from the peptide and are thus crucial for peptide specificities [76]. The presence of these well-defined anchor residues and anchoring grooves, which provide stability necessary for allele-specific recognition, helps to explain why each allelic form of class I molecule binds a diverse yet specified spectrum of peptides [77].

Before the CGMD simulation, we prepared the initial conformation of the peptide-MHC complex using protein-peptide docking. However, we identified at least three major challenges to protein-peptide docking: (i) modeling significant conformational changes of both peptide and protein molecules (flexibility problem); (ii) selection of the highest accuracy structure out of many generated models (scoring problem); and (iii) integration of experimental data and computational predictions into the protein–peptide docking scheme (integrative modeling) [78]. Interaction details between peptide epitopes and MHC-I molecules have been known, where the peptides anchor to the MHC-I binding pockets through their second and last residues. Such knowledge benefits us from employing the template-based docking method that utilizes homology modeling to prepare peptide-MHC (pMHC) complexes studied in this work. A similar approach has also been adopted by PANDORA, a technique to model pHMC with a success rate of 93% [79].

The peptide-MHC overall interactions are notably contributed by the binding of primary anchor residue-2 (P2) with the highly hydrophobic B pocket and residue-9 (P9) with the F pocket carrying neutral charges. These findings are consistent with previous studies that showed the P2 and P9 or PΩ as the primary anchor of HLA-A*02:01 [80,81]. Hence, the C-terminal did not allow any charged amino acids. The P2 in the strong-binder peptides are Isoleucine and Leucine (F**I**AGLIAIV–Y**L**QPRTFLL), the branched hydrophobic amino acids, and therefore it fits naturally with the B pocket in MHC class I. On the other hand, in weak-binder (V**Y**DPLQPEL), there is Tryptophan, the large aromatic amino acid with an indole ring that fits less. Also, the polar amino acids Glutamine (E**Q**DKNTQEV) and Asparagine (T**N**GTKRFDN) in weak- and non-binder peptides are incapable of binding to the B pocket. Furthermore, the P9 in strong- and weak-binder peptides (FIAGLIAI**V**—YLQPRTFL**L** and EQDKNTQE**V**—VYDPLQPE**L**) are uncharged and have a good affinity towards the F pocket by promoting hydrophobic interactions. Whereas in non-binder TNGTKRFD**N**, the P9 is asparagine with a positive charge. Based on the interaction data (Fig 6A–6D), we can observe that the central region of peptides rarely involves in the binding with MHC. This finding agrees with the previous work by Szeto et al., which observed YLQPRTFLL, the same one that was included in the strong-binder category in our study. It is observed that P4 –P5 and P8 interact with CDR1α, CDR3α, and CDR1-3β in TCR [73]. The N-terminal location for TCR

binding may be inherently suboptimal, resulting in low affinities and encouraging escape from negative selection. On the other hand, a higher affinity can be obtained by focusing on the central or occasionally C-terminal region of peptides, which are probably more suitable locations for TCR binding [50,82,83]. The interaction energy of P1 in the N-terminal is observed to be the highest, which is consistent with the other CG simulation of peptide detachment from the MHC. It is found that the detachment phases typically begin C-terminally, with the N-terminal end following later and more slowly [84].

About the measurement method of T-cell response titer, there are several ways that can be used. One method is the enzyme-linked immunospot (ELISPOT) assay, which measures the number of cytokine-producing T cells in response to a specific antigen [85]. Another method is flow cytometry, which can be used to quantify the number of T cells expressing specific surface markers or intracellular cytokines [86]. Additionally, the enzyme-linked immunosorbent assay (ELISA) can be used to measure the concentration of specific cytokines or other molecules produced by T cells [87]. These methods provide quantitative measurements of T cell response titer and can be used to assess the magnitude and quality of T cell responses in various contexts, such as infectious diseases, autoimmune disorders, and cancer immunotherapy. Moreover, the interferon test, commonly known as IGRA, has been widely used to measure T-cell responses for latent tuberculosis infection [88,89] and Covid-19 diagnostic test [90,91]. Our findings enable vaccine designers to utilize molecular dynamics simulations for definitive binding category classification. The established concept from this study holds the potential for subsequent refinement, thereby augmenting the success rate of authentic epitope discovery in vaccine development.

## Acknowledgments

The authors would like to thank the Research Center for Molecular Biotechnology and Bioinformatics–Universitas Padjadjaran for providing a computing research facility.

## Author Contributions

**Conceptualization:** Muhammad Yusuf, Ari Hardianto, Neni Nurainy, Acep Riza Wijayadikusumah, Ines Irene Caterina Atmosukarto, Toto Subroto.

**Data curation:** Muhammad Yusuf, Wahyu Widayat, Yosua Yosua, Angelica Shalfani Tanudireja, Farhan Azhwin Maulana, Umi Baroroh.

**Formal analysis:** Muhammad Yusuf, Wanda Destiarani, Wahyu Widayat, Yosua Yosua, Gilang Gumilar, Angelica Shalfani Tanudireja, Fauzian Giansyah Rohmatulloh, Farhan Azhwin Maulana, Umi Baroroh, Ari Hardianto, Rani Maharani.

**Funding acquisition:** Muhammad Yusuf.

**Investigation:** Muhammad Yusuf, Wahyu Widayat, Yosua Yosua, Fauzian Giansyah Rohmatulloh, Farhan Azhwin Maulana, Ari Hardianto.

**Methodology:** Muhammad Yusuf, Wanda Destiarani, Yosua Yosua, Gilang Gumilar, Ari Hardianto, Rani Maharani, Neni Nurainy, Ines Irene Caterina Atmosukarto.

**Resources:** Muhammad Yusuf, Gilang Gumilar, Neni Nurainy, Acep Riza Wijayadikusumah, Ryan B. Ristandi, Ines Irene Caterina Atmosukarto, Toto Subroto.

**Software:** Fauzian Giansyah Rohmatulloh.

**Supervision:** Muhammad Yusuf, Rani Maharani, Neni Nurainy, Acep Riza Wijayadikusumah, Ryan B. Ristandi, Ines Irene Caterina Atmosukarto, Toto Subroto.

**Visualization:** Wanda Destiarani, Umi Baroroh, Ari Hardianto.

**Writing – original draft:** Muhammad Yusuf, Wanda Destiarani.

**Writing – review & editing:** Muhammad Yusuf, Wanda Destiarani, Ari Hardianto.

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
