## [Decision Letter · Decision Letter 0]

20 Aug 2023

PONE-D-23-23650Coarse-Grained Molecular Dynamics-guided immunoinformatics to explain the binder and nonbinder classification of cytotoxic T-cell epitope for SARS-CoV-2 peptide-based vaccine discoveryPLOS ONE

Dear Dr. Yusuf,

Thank you for submitting your manuscript to PLOS ONE. After careful consideration, we feel that it has merit but does not fully meet PLOS ONE’s publication criteria as it currently stands. Therefore, we invite you to submit a revised version of the manuscript that addresses the points raised during the review process.

We look forward to receiving your revised manuscript.

Kind regards,

Sheikh Arslan Sehgal, PhD

Academic Editor

PLOS ONE

Journal Requirements:

4. Please note that funding information should not appear in any section or other areas of your manuscript. We will only publish funding information present in the Funding Statement section of the online submission form. Please remove any funding-related text from the manuscript.

5. Please amend the manuscript submission data (via Edit Submission) to include author Ari Hardianto.

Reviewers' comments:

Reviewer's Responses to Questions

**Comments to the Author**

1. Is the manuscript technically sound, and do the data support the conclusions?

Reviewer #1: Yes

2. Has the statistical analysis been performed appropriately and rigorously? 

Reviewer #1: Yes

3. Have the authors made all data underlying the findings in their manuscript fully available?

Reviewer #1: Yes

4. Is the manuscript presented in an intelligible fashion and written in standard English?

Reviewer #1: Yes

5. Review Comments to the Author

Reviewer #1: General comment:

The computational technique elicited in this paper deserve attention as it could produce vaccine designs while overcoming the troubles in time and space complexity. However, there are some glitches in the biological background that should be clarified in the specific comments part.

Specific comment:

1. It is true that the choice of T-cell-based epitopes may confer long-term protection against infection. However, short-term protection can't be neglected, and it is still the most viable way to ensure vaccine efficacy in clinical setting. B-cell response may be weak, but it is much easier to measure with simple biomedical lab setting. Determining vaccine efficacy from T-cell titers will possibly need NGS machine, and not all hospital and clinics have that. You need to narrate a stronger argument both in discussion and intro, why should the B-cell epitopes be neglected here. Because all existing vaccine in the market, whether it is live attenuated, mRNA, recombinant, and others, worked by inducing the B-cell response and they have been proven to slow down, if not to stop, the progression of COVID-19 pandemics. Although it is not your forte, you need to think or predict how to measure the vaccine efficacy later on.

2. What is the gap between the available vaccine design with yours? It seemed that there are already a lot of COVID-19 epitopes-based vaccine design, and some of them combining both T-cell and B-cell epitopes. So this kind of design could potentially confer both long and short-term protections, and could be measured with its antibody titers only. Please kindly check this link in PUBMED, and kindly justify the available gap! https://pubmed.ncbi.nlm.nih.gov/?term=T-cell+epitope+sars-cov-2+vaccine

3. Pertaining your methodology, did you incorporate the latest finding of omicron subvariant (XBB)? If you don't, you need to declare this condition as limitation of your study.

4. Only after reading thoroughly your whole manuscript, I can understand that you invoke molecular docking method for ensuring the formation of protein-peptide complex. However, it is not elicited in your methodology section. Please kindly create a specific 'molecular docking' subsection there, so the reader can have better insight on your invoked methods.

6. PLOS authors have the option to publish the peer review history of their article (what does this mean?). If published, this will include your full peer review and any attached files.

Reviewer #1: No

---

## [Author Response · Author response to Decision Letter 0]

8 Sep 2023

To Editor:

Dear Prof. Emily Chenette,

We have revised the manuscript according to the Reviewer’s comments and to the PLOS ONE guidelines. We also separate the title page and the main manuscript into different files. Regarding the copyedit for language, we checked the manuscript's language usage, spelling, and grammar using the premium version of the Grammarly program. Hopefully, it can ensure the contents are grammatically correct. In addition, The Data Availability statement has been provided in the Data Availability section at the bottom of the manuscript. The URL for each data that has been used is also written. The funding statement in the Acknowledgements section has been removed. About the authorship, we have added Ari Hardianto to the submission system. Thank you very much.

To Reviewer:

Dear Prof.,

We would like to thank you for having your time to review our manuscript. Your critical and positive suggestions really help us to improve the manuscript substantially. We hope that the current state of the manuscript addresses all of your concerns and fulfill the requirement to be accepted in PLOS One. Please find our point-by-point responses to your comments in the attached PDF file. Thank you very much.

Kind regards,

Muhammad Yusuf

---

## [Editor Report · Decision Letter 1]

13 Sep 2023

Coarse-Grained Molecular Dynamics-guided immunoinformatics to explain the binder and nonbinder classification of cytotoxic T-cell epitope for SARS-CoV-2 peptide-based vaccine discovery

PONE-D-23-23650R1

Dear Dr. Yusuf,

We’re pleased to inform you that your manuscript has been judged scientifically suitable for publication and will be formally accepted for publication once it meets all outstanding technical requirements.

Kind regards,

Sheikh Arslan Sehgal, PhD

Academic Editor

PLOS ONE
---

## [Editor Report · Acceptance letter]

27 Sep 2023

PONE-D-23-23650R1 

Coarse-Grained Molecular Dynamics-guided Immunoinformatics to Explain the Binder and Non-binder Classification of Cytotoxic T-cell Epitope for SARS-CoV-2 Peptide-based Vaccine Discovery 

Dear Dr. Yusuf:

I'm pleased to inform you that your manuscript has been deemed suitable for publication in PLOS ONE. Congratulations! Your manuscript is now with our production department. 

Kind regards, 

on behalf of

Dr Sheikh Arslan Sehgal 

Academic Editor

PLOS ONE